# Potential of Bitter Medicinal Plants: A Review of Flavor Physiology

**DOI:** 10.3390/ph17060722

**Published:** 2024-06-03

**Authors:** Germán Zuluaga

**Affiliations:** 1Grupo de Estudios en Sistemas Tradicionales de Salud, Escuela de Medicina y Ciencias de la Salud, Universidad del Rosario, Bogotá 111711, Colombia; gzuluaga@cemi.org.co; Tel.: +57-311-2179102; 2Centro de Estudios Médicos Interculturales, Cota 250010, Colombia

**Keywords:** physiology of flavor, medicinal bitterness, traditional medicine, medicinal plants

## Abstract

The function of the sense of taste is usually confined to the ability to perceive the flavor of food to assess and use the nutrients necessary for healthy survival and to discard those that may be harmful, toxic, or unpleasant. It is almost unanimously agreed that the perception of bitter taste prevents the consumption of toxins from plants, decaying foods, and drugs. Forty years ago, while practicing medicine in a rural area of the Colombian Amazon, I had an unexpected encounter with the Inga Indians. I faced the challenge of accepting that their traditional medicine was effective and that the medicinal plants they used had a real therapeutic effect. Wanting to follow a process of learning about medicinal plants on their terms, I found that, for them, the taste of plants is a primary and fundamental key to understanding their functioning. One of the most exciting results was discovering the therapeutic value of bitter plants. The present review aims to understand whether there is any scientific support for this hypothesis from the traditional world. Can the taste of plants explain their possible therapeutic benefit? In the last 20 years, we have made novel advances in the knowledge of the physiology of taste. Our purpose will be to explore these scientific advances to determine if the bitter taste of medicinal plants benefits human health.

## 1. Introduction

The function of the sense of taste is usually confined exclusively to the ability to perceive the flavor of food to assess and use the nutrients necessary for healthy survival and to discard those that may be harmful, toxic, or unpleasant.

For many years, four primordial or primary tastes were identified: sweet, salty, sour, and bitter, the combination of which gives rise to the immense mosaic of secondary tastes that we can perceive. Since 1909, Japanese professor Ikeda indicated the existence of another taste modality he called umami, which translates to “tasty”, “delicious”, or “exquisite” [1]. Currently, most research considers five taste modalities, including umami [2].

There is some unanimity about the biological role of each of the flavors, as summarized by Chandrashekar et al.:

“Sweet taste enables the identification of energy-rich nutrients, umami enables the recognition of amino acids, salty taste ensures the proper balance of dietary electrolytes, and sour and bitter tastes warn about the ingestion of potentially harmful and/or poisonous chemicals. In humans, taste has the additional value of contributing to the overall pleasure and enjoyment of a meal. Surprisingly, although we can taste a wide range of chemical entities, it is now generally accepted that, qualitatively, they evoke few distinct taste sensations: sweet, bitter, sour, salty and umami. Although this repertoire may seem modest, it has satisfied the evolutionary need for an effective and reliable platform to help recognize and distinguish major dietary components.” [3] (p. 288)

In other words, these flavors act synergistically to mediate appetite-protective responses, such as energy regulation, and the intake of salt and protein. They also warn against the consumption of toxic substances and in some cases determine our food preferences [2].

In particular, it is almost unanimously agreed that the perception of bitter taste has the function of preventing the consumption of toxins from plants, decaying foods, and drugs, consistently producing rejection or aversion responses [4]. Similarly, most authors agree that the function of sour taste is to prevent the ingestion of potentially harmful, poisonous, or simply unpleasant chemicals. Indeed, the ability to sense sourness provides an important sensory signal to avoid ingestion of unripe, spoiled, or fermented foods. Taste receptors in the oral cavity trigger aversive behaviors in response to acidic stimuli [5]. Most people respond to the consumption of lemon juice with a violent contortion of the face that clearly indicates a rejection of the sour taste [6].

It does not seem to be disputed that the function of bitter and acidic substances is exclusively to prevent the ingestion of toxic, poisonous, or unpleasant substances. Such a perspective has given rise to the widespread notion that bitter and acidic substances should be avoided. This current scientific knowledge promotes a rejection of anything bitter or acidic, which coincides with a trend that has become widespread in the eating habits of the modern world: acidophobia and bitterphobia, understandable because of the usual initial rejection or aversive behavior that both tastes tend to produce. We can cite a few examples: hardly anyone wants to consume stimulating beverages or foods such as coffee, tea, and chocolate without adding sugar; citrus fruits, which are usually acidic in their natural form, are replaced by genetic varieties that intentionally seek to reduce their acidity and increase their sweetness, as is the case with the new grafted varieties of oranges and lemons; refreshing and thirst quenching drinks offered by the beverage industry are usually nuanced with sugar, sweeteners, flavorings, and fizzy substances. Moreover, most people are abandoning the consumption of medicinal plants, leaving aside mainly bitter plants, and only keeping those that are considered sweet or aromatic. Undoubtedly, the diet of the modern world favors foods and beverages with sweet and salty flavors, abandoning more and more the consumption of substances with bitter and sour flavors. Does this phenomenon of change in the diet have any consequences for human health?

Forty years ago, while I was practicing medicine in a rural area of the Colombian Amazon, I had an unexpected encounter with the Inga Indians. I was faced with the challenge of accepting that their traditional medicine was effective and that the medicinal plants they used had a real therapeutic effect. Obviously, I wanted to approach the study of medicinal plants under the terms and categories of Western medicine, looking for plants with anti-inflammatory, antihypertensive, and hypoglycemic effects and plants for arthritis, collagen diseases, or cancer, among others. And, of course, based on the criteria of phytochemistry and classical pharmacological research, I wanted to find the active principles that would explain the possible therapeutic effect of medicinal plants.

Although this scientific path for the study of medicinal plants continues to be the standard, both in botany and ethnobotany, as well as in phytochemistry and phytopharmacology, our experience has shown us that traditional systems of knowledge respond to other epistemological paths, other methods, other ways of knowing and using plants. To begin with, we were forced to try to understand and interpret the language used by indigenous healers. None of the terms used by our medicine and our western science correspond to the terms used by them. While, for example, I asked for plants for systemic lupus erythematosus or diabetes, they answered with hot or cold plants, purgative or depurative plants, plants for bad air, or for fear and fright. There is, thus, a hermeneutic challenge here that requires us to be willing to engage in an authentic dialogue of sciences.

Accepting this new disposition and wanting to follow a process of learning about medicinal plants, in their own terms, in addition to the predominant category of hot and cold plants (a concept that has been scorned even by anthropologists, ethnobotanists, and pharmacologists), we found that, for them, the taste of plants is a primary and fundamental key to understanding their functioning. Thus, a new ethno-classification of medicinal plants was opened: sweet or aromatic, fresh, acid, and bitter. According to them, flavor is the key to their function and efficacy. During the following years, our studies have tried to follow this new epistemological track. One of the most interesting results was the discovery of the therapeutic value of bitter plants. Indeed, bitter became a key to understand the universe of indigenous and traditional therapeutics [7]. As a result of this experience, we have even come, as physicians, to recommend the consumption of bitter plants for preventive and therapeutic purposes [8]. Although clinical experience has shown its benefits, we are still far from being able to confirm it with experimental epidemiological studies.

As Trius and Moreno state: “Although numerous bitter medicinal plants have been used in folk and traditional medicine and cuisine and inherited by generations, the scientific integration of modern medicine and traditional (ancient) wisdom remain limited.” [9]. Indeed, recent ethnobotanical studies report the bitter properties of some medicinal plants. However, they do not usually give the traditional explanation of their efficacy or are associated with benefits related to events such as fever, inflammation, liver, chest diseases, edema, diarrhea, digestion, malaria, and intestinal parasitosis, among others. In Table 1, I cite some bitter medicinal plants reported to be used in cosmopolitan or local communities. Although phytochemical and pharmacological studies have been conducted on most of them, it is rarely reported whether the active ingredients with a physiological or therapeutic effect have a bitter taste themselves.

The objective of the present review is to understand whether there is any scientific support for this hypothesis from the traditional world. Can the taste of plants really explain their possible therapeutic benefit? Is bitter taste, so little appreciated in our modern world, just an expression of the toxic effect of substances, plants, and foods, or does it have any importance for human health?

In the last 20 years, we have found novel advances in the knowledge of the physiology of taste, for example in the discovery of cell membrane receptors that explain in a fascinating way the way in which these substances activate and generate the chemical and electrical activities involved in the physiology of taste, and in the functional complexity of taste cells, buds and papillae, ions and organic molecules, neurotransmitters and routes or connections of nerve bundles with the different structures of the brain that allow their emission and final interpretation in the cerebral cortex.

Our purpose will be to explore these new scientific advances to determine if the bitter taste of medicinal plants has benefits for human health.

## 2. The Sense of Taste and Flavor

Taste is recognized as one of the fundamental senses possessed by living beings, thanks to which they can perceive and interact with the outside world. Its importance is recognized even in unicellular and prokaryotic organisms, since it allows them to assess the hydroelectrolytic conditions of the environment and thus obtain nutritive substances or repel those that may be harmful.

The structure and physiology of taste has evolved in the animal kingdom, reaching increasingly complex and well-developed mechanisms in vertebrate animals, mammals, and humans. Whereas in invertebrate organisms the cells responsible for taste sensing are primitive neurons with ciliated dendrites, as identified in the nematode *Caenorhabditis elegans* and the fruit fly *Drosophila melanogaster*, vertebrate taste receptor cells are not neurons but originate in the epithelial covering of the body [20].

In mammals, the taste system is located mainly in the tongue and oral mucosa, from where it is transmitted to the brain via cranial nerves, following a complex route until taste perception is recognized by the cerebral cortex. In simple terms, food chemicals dissolve in saliva and come into contact with taste cells through the taste pore. There, they interact with taste receptors, triggering electrical changes in taste cells, which stimulate the emission of chemical signals, an activity that is translated into impulses sent to the brain [21].

But the sense of taste is much more complex than just receptors for basic flavors and the chemical interactions they generate in taste cells. Despite the tendency to identify taste information in terms of five modalities, the taste system also interprets other attributes derived from chemical stimuli. An intense taste can be pleasant, unpleasant, or neutral. Neurons in the taste pathway register these attributes simultaneously, like the visual system represents shape, brightness, color, and motion. Very often, taste receptors also respond to tactile and thermal stimuli [21].

For this reason, it is important to distinguish between taste and flavor. Taste refers to the capacity of living organisms, from invertebrates to mammals and humans, to perceive an intrinsic biological property of many substances of mineral, vegetable, and animal origin. On the other hand, the final perception of flavor is the result of the activation of several sensory systems: taste, smell, tactile, and thermal sensitivity, among others [22]. Flavor is, therefore, a complex amalgam of sensory information provided by taste, smell, and the tactile sensation of food when it is chewed, a characteristic that scholars usually refer to as mouthfeel. Thus, the word taste is applied, strictly speaking, only to sensations coming from the specialized cells of the mouth, although taste and flavor are commonly used interchangeably [21]. This distinction is also evident in the Spanish language: “Taste (*gusto*) is commonly confused with flavor (*sabor*), the combined sensory experience of olfaction and gustation.” [23].

## 3. The Taste System

Although the physiology of taste has been known with some precision since the beginning of the 20th century, in recent decades, thanks to scientific advances in genetics and immunology, new and more detailed knowledge has been obtained that allows for greater understanding, although many concerns remain to be resolved.

From the phylogenetic point of view, the basic unit of the sense of taste is based on ciliary structures that, from unicellular organisms onwards, develop in certain cells [24], which consequently acquire their specialized character and for this reason are known as taste cells or taste receptor cells (TRCs). In vertebrates, they are small bipolar cells with ciliary properties [2], which have two functionally important connections: microvilli in contact with the oral cavity and synapses with sensory nerve fibers [20].

Taste cells may be alone or in groups of 50 to 100 cells, thus forming the functional units known as taste buds [2]. In each taste bud, there are four morphologically distinguishable cell types. On the one hand, there are the Type IV basal cells confined to the basal surface, which are not involved in taste physiology, serve as support, and are considered progenitors, since they are in charge of cell turnover. On the other hand, there are the three cell groups involved in taste physiology: Type I, Type II, and Type III cells [25] (see Figure 1).

Taste buds are in turn grouped into larger structures, known as taste papillae, distributed in different geographical areas of the tongue and in some points of the oral mucosa. Four types are distinguished: filiform, fungiform, caliciform, and foliated [26].

Finally, taste papillae are innervated, in a differentiated and apparently specific way for each group of papillae, by nerve bundles and fibers of the cranial nerves, especially the trigeminal (V), facial (VII), glossopharyngeal (IX), and pneumogastric (X) [26], through which gustative sensitivity of the tongue is conveyed to the central nervous system, entering the brain stem, following the hypothalamus, to reach the frontal lobe of the cerebral cortex [27].

## 4. The Physiology of Taste

The physiological basis of all senses, such as taste, smell, hearing, balance, touch, and vision, has a common biochemical conceptual framework, based on the depolarization of the cell membrane, leading to an ionic exchange that in turn generates electrical and chemical changes inside the cell. However, the initial trigger is different from case to case. This is how the three known mechanisms are described: (a) mechanotransduction, in which the trigger is a pressure change, as occurs in the cochlear and vestibular receptors of the organ of hearing, (b) phototransduction, in which the trigger is light, as occurs in the retinal receptors of the organ of the eye, and (c) chemotransduction, in which the trigger is a chemical substance or molecule, as occurs in the gustatory receptors of the organ of taste [24]. Some authors add a fourth mechanism, thermoreception, in which temperature is the trigger, perhaps also present in the sense of taste [28]. The physiology of taste is mainly chemosensory.

Like neurons, taste cells present at rest a net negative charge inside and a net positive charge outside. The chemicals in the food modify this situation through various mechanisms that increase the concentration of positive ions inside. The result is the suppression of the charge difference between the outside and the inside of these cells. This depolarization causes the taste cells to release neurotransmitters to the exterior—molecules that trigger the transmission of electrical messages in the neurons in contact [21]. For this depolarization to be possible, taste cells have receptor proteins or cell surface receptors in their membranes, which can be of two kinds:(a)Channel or ionotropic receptors. They are known by the acronym TRP, which stands for transient receptor potential. In turn, according to their molecular specificity, they are subclassified into different groups: TRPc, TRPm, TRPp, TRPv, among others [29].(b)G protein-associated or metabotropic receptors [30]. They are known by the acronym GPCR (G protein-coupled receptor, GPCRs for the plural) [31].

Based on the receptor protein, we classify taste cells as ionotropic, which are those in which the protein per se is an ion channel, and metabotropic, in which the receptor is associated with a G protein [2]. Both taste receptor proteins are mounted on the microvilli, acting as molecular antennae that listen to the chemical environment. Upon binding to molecules contained in food, the receptors trigger a signaling cascade within the cells, the end result of which is manifested in the opening and closing of other intracellular ion channels [21], activating synapses and thus causing excitation of nerve fibers [20]

### 4.1. Channel or Ionotropic Receptors

Channel receptors, which are poreiform proteins and are an ion channel per se, correspond to a large family of receptors, called TRP, whose function involves from insect sense organs to that of humans, including the functions of calcium and magnesium homeostasis, vasoregulation, gastrointestinal motility and hepatic development, among many others [24]. From the phylogenetic point of view, they are considered more primitive and less complex.

TRP channels are thus an ancient sensory apparatus of the cell and have been adapted to respond to all kinds of stimuli, both from inside and outside the cell. It is even considered that almost all cells have these channel receptors. They are thus the forefront of our sensory systems, responding to temperature, touch, pain, osmolarity, pheromones, taste, and other stimuli. The known functions are diverse. Yeasts use a TRP channel to sense and respond to hypertonicity. Nematodes use TRP channels at the tips of neuronal dendrites to sense and avoid noxious chemicals. Male mice use a pheromone-sensing TRP channel to distinguish males from females. Humans use TRP channels to appreciate flavors and to discriminate heat and cold [29]. Nearly all mammalian TRP channel genes are now known [32].

Epithelial sodium channels (ENaCs) were the first to be recognized in taste physiology [33]. Indeed, this new class of channels, the epithelial sodium channel (ENaC)/degenerin gene family, was discovered in the 1990s. Different investigations confirmed the existence of these channels in the African frog *Xenopus laevis*, in the mollusk *Helix aspersa,* and in the nematode *Caenorhabditis elegans*, with great genetic sequence homology between them and also coinciding with the recently described ENaCs in mammals and humans [34].

ENaCs were initially recognized for their specific response to amiloride, a potassium-sparing diuretic approved in 1967 for the treatment of arterial hypertension and heart failure, because they played a crucial role in the uptake of sodium (Na^+^) in the distal part of the renal tubule, which was the target of aldosterone action. Hence, they were named epithelial sodium channels (ENaCs). Their homology with the degenerin channel of the nematode *C. elegans* led to the current classification of the epithelial sodium channel (ENaC)/degenerin gene family [34].

We now know that in humans, ENaCs are found in epithelial cells of the kidney, lung, respiratory tract, male and female reproductive tracts, sweat and salivary glands, placenta, colon, and some other organs. In these epithelia, ENaCs allow the flow of Na^+^ ions from the extracellular fluid into the epithelial cell. As Na^+^ is one of the major electrolytes in the extracellular fluid, the change in osmolarity initiated by Na^+^ flux is accompanied by a flux of water accompanying Na^+^ ions. Thus, the ENaC has a central role in extra- and intracellular fluid volume regulation, electrolyte homeostasis, blood pressure, airway clearance, germ cell transport, fertilization, implantation, and cell migration [33].

Observations that amiloride and other ENaC blockers alter taste responses to sodium salts led to the hypothesis that the ENaC was involved in salt taste reception. Finally, one study in mice showed that the ENaC responds to detecting sodium taste [35] and is primarily responsible for the salty taste.

New research showed that also other TRP channels are involved in thermosensation, mechanosensation, odor, and taste. For example, TRPm5 was found to be abundantly expressed in taste receptor cells, where it is involved in the perception of sweet, sour, and bitter tastes [36]. The TRPp3 channel receptor was also identified to be associated with the perception of sour taste [37], and two members of the PKD-like TRP ion channel may function as sour taste receptors [38]. For their part, Zhang et al. demonstrated that an ion channel called Otopetrin-1, a proton-selective channel normally involved in the sensation of gravity in the vestibular system, is essential for the perception of sourness in the gustative system [5].

In summary, ion channels or Transient Receptor Potential (TRP) channels are involved in taste physiology: (a) epithelial Na^+^ channels (ENaC) are mainly responsible for a salty taste, (b) TRPm5 channels are involved in the perception of sweet, bitter, and sour tastes, and (c) acid-sensitive ion channels (ASICs), PKD-like TRP channels, and an ion channel called Otopetrin-1 are responsible for sour taste (see Figure 2).

### 4.2. G Protein-Coupled Receptors and G Protein

G protein-coupled receptors (GPCRs) are the largest known family of membrane receptors in eukaryotes, consisting of more than 800 members. These receptors recognize a wide range of extracellular stimuli such as photons, hormones, neurotransmitters, tastes, and odors to induce intracellular signaling pathways [31]. Because of their structure, forming seven transmembrane α-helices, they are also called receptors of the seven transmembrane domains [39]. More than 100 members of this family have been described in mammalian cells [30], and they constitute the third largest gene family in humans [40]. More than 50% of commercially available drugs directly or indirectly target GPCRs, highlighting their importance in various biological and therapeutic mechanisms [31].

The American Robert Lefkowitz, since the 1960s, has postulated the existence of receptors that formed part of the cell signaling system, based on his studies on alpha- and beta-adrenergic receptors. In 1980, Lefkowitz and his colleagues De Lean and Stadel proposed the so-called “ternary complex model”, which involved the interaction of the receptor with a third component, which generated a receptor with high affinity for the ligand. It was later discovered that this third component was an intracellular G protein. Shortly thereafter, with fellow American Brian Kobilka, they succeeded in isolating the gene, making it clear that the beta-adrenergic receptor was part of a large family of receptors that had common features in terms of their structure and the way they functioned. In 2011, Kobilka, with his own research group, succeeded in obtaining the structure of the beta-adrenergic receptor bound to a G protein by X-ray crystallography [41]. Finally, both researchers received the Nobel Prize in Chemistry in 2012 for “their studies on G protein-coupled receptors” [40].

The researchers Alfred Gilman and Martin Rodbell, when studying cell stimulation by epinephrine, found that, upon binding to the receptor, the hormone did not stimulate enzymes directly, but did so through proteins coupled to the receptor, which were responsible for the activation of guanine triphosphate (GTP), which is why they called them G proteins. These stimulate adenylyl cyclase, which in turn produce cyclic adenosine monophosphate (cAMP) as a second messenger [42]. Thus, upon discovering the role of these proteins in cellular signal transduction, they received the Nobel Prize in 1994 [39].

In relation to the physiology of taste, in 1992, researchers McLaughlin and McKinnon collaborated with Margolskee to identify a protein they named gustducin, given its molecular similarity to transducin, another retinal cell protein that helps transform or transduce the light signal reaching the retina into an electrical impulse constitutive of vision. Gustducin and transducin are G proteins that are bound to the inner side of their respective surface receptors. When a genuine taste molecule binds to a receptor taste cell, with the specificity of a key in its lock, the gustducin subunits (α, β and γ) separate and catalyze biochemical reactions that result in the opening or closing of ion channels [21]. Gustducin has also been detected in nasal mucosal and stomach cells [20].

Two groups of researchers, the first led by Charles Zuker at the Howard Hughes Institute of Medicine at the University of California at San Diego and Nicholas Ryba at the National Dental and Craniofacial Institute, and the second led by Linda Buck at Harvard Medical School, identified in 1996 in mice and humans the true bitter taste receptors that activate gustducin. Both teams found that bitter taste receptors (T2Rs for plural and T2R for singular) belonged to a family of similar receptors consisting of perhaps 40 to 80 members [21].

Very soon, the receptors that recognize sweet substances and the umami receptor were also sought, and it was discovered that they were proteins that shared similar amino acid sequences and were named sweet taste receptors (T1R) [43]. The genomic library research failed to find additional members, so it was finally determined that two G protein-coupled receptors were involved in taste physiology: T1R, responsible for identifying a sweet taste and perhaps umami, and T2R, responsible for identifying a bitter taste [44].

The next and almost simultaneous task was to identify the encoding genes for these receptor proteins that were finally discovered in 2000, based on analyses of recently published human genome sequences in the genome regions related to bitter taste repotentiation in humans and mice [45]. Adler et al. examined a region of human chromosome 5 linked to the perception of a bitter compound, 6-n-propyl-2-thiouracil (PROP) [46], and by 2006 the genetic map of the T1R and T2R receptors was fully established [47].

As we have seen, different routes have been taken to obtain knowledge of the membrane receptors necessary to understand the chemosensory physiology of taste, in particular, the receptors GPCR (and their specific G protein, gustducin) and TRP. There is already a consensus that the somewhat more complex sweet, umami, and bitter tastes are mediated by GPCR (T1R and T2R) that in turn stimulate the protein gustducin [47], whereas the somewhat simpler salty and sour tastes are mediated by TRP [36].

### 4.3. Taste in the Central Nervous System

The neuronal transmission of taste perception reaches the central nervous system by means of nerve bundles and fascicles through the cranial nerves, activating the Nucleus of the Solitary Tract in the brain stem, responsible for the initial processing of information. Following the central route, information reaches the Parabrachial Nucleus, a more complex structure that integrates various aspects of sensory information. From there, information reaches the Thalamus, the third structure activated in the taste pathway, where the essential coding of the five basic tastes is integrated with other sensory stimuli, such as smell and texture. Finally, the information reaches the taste cortex, located at the boundary between the anterior insula and the frontal operculum of the cerebral cortex [48], a multimodal area capable of integrating all aspects of information, since its neurons not only respond to taste but also to temperature, touch, pain, and the state of the viscera (see Figure 3). The response orchestrated by groups of neurons in this cortex, called “ensembles”, enables processing and integrating all the characteristics contained in a food and comparing them with the previously stored information associated with that taste. In this way, a specific combined code is achieved that will allow the subsequent recognition of that flavor within a wide range of combinations [27].

One area of intense research is how taste cues are encoded by the nervous system. The principles of sensory coding from the retina to the visual cortex were elucidated decades ago. We have a thorough understanding of the tonotopic and computational maps of the auditory system. Lateral inhibition and somatosensory receptive fields are well defined. Knowledge about taste is lacking and we still do not understand how the brain distinguishes sweet, sour, salty, etc. If taste does not follow a simple label code, how are taste signals transmitted and decoded? Studies currently underway shed light on the possibility that taste is encoded in the time domain, i.e., by the frequency and pattern of action potentials in the hindbrain and cortical neurons. Other laboratories are exploring higher-order cortical processing using functional magnetic resonance imaging to address the interplay between taste detection, preference, and appetite regulation [23].

It is now known that neurons in the different areas of the taste pathway can modify their chemical and electrical activity and their conformation, depending on the type of experience associated with taste. Thus, the taste cortex neurons are highly “plastic”, i.e., they adapt rapidly and change their response as the hedonic value is transformed during the consumption of a new taste, but also during the consumption of a familiar taste, which allows for the constant updating of taste memory over days, months, or years later. Thanks to this plasticity, we can initially recognize that something is bitter in order to learn that it is not harmful and, subsequently, this now familiar taste can become pleasant with a highly pleasurable value. Classic examples are beer, wines, and some strong cheeses, which once tasted and once the initial aversion is overcome, usually provoke a pleasurable increase in their consumption. It is true that every time we consume a particular food, we can recognize it specifically, even though its hedonic connotation changes from time to time. In other words, the ability to recognize taste modalities remains intact in our brain; foods always interact with the same receptors in the mouth, but these end up activating different patterns of neurons depending on the history of experiences we have with that taste [27].

## 5. The Bitter Taste

It is produced almost entirely by organic substances, rather than ionic chemical elements. Initially it was thought that only two types of substances produced this taste: alkaloids and long-chain nitrogen-containing organic substances [26]. But in reality, thousands of compounds are perceived as bitter by humans. There is no reliable inventory in nature of bitter molecules but tens of thousands are estimated because, for naturally occurring bitter compounds alone, about 10% of plant species can synthesize toxic secondary metabolites and 2500 plant species alone contain cyanogenic glycosides, most of which are bitter. Bitter compounds are not only numerous but also structurally diverse. They include, among others, fatty acids, peptides, amino acids, amines, amides, azacycloalkanes, N-heterocyclic compounds, ureas, thioureas, carbamides, esters, lactones, carbonyl compounds, phenols, crown ethers, terpenoids, secoiridoids, alkaloids, glycosides, flavonoids, steroids, halogenated or acetylated sugars, and metal ions [49]. Of these, quinine is used as the standard reference for expressing a bitter taste [26].

Many bitter substances are of genuine plant origin, but others are derived from animals or are generated during food processing, aging, or decomposition. Culinary Maillard reaction and fermentation processes are also rich sources of bitter compounds, while chemical synthesis has provided unique bitter chemotypes [49].

As already mentioned, it is almost unanimous and universally accepted that the perception of bitter taste prevents the consumption of toxins from plants, decaying foods, and drugs, consistently producing rejection or aversion responses [4]. Such a claim has been made since the end of the last century and persists to this day:

“The primitive function of this taste quality would seem to be to provoke the rejection of toxic substances.” [50]

“Bitter taste perception provides animals with critical protection against ingestion of poisonous compounds.” [45]

“Taste is a different matter, especially where bitter compounds are concerned. Virtually every naturally occurring toxin tastes bitter, ‘so bitterness clearly evolved with the sole purpose of warning you against the ingestion of toxic substances,’ says Zuker. The important thing is to recognize and reject anything bitter, not to get hung up on distinctions among different compounds. Indeed, experimental evidence indicates that humans are unable to discriminate one bitter substance from another.”[51]

“Bitter taste effectively warns us not to ingest potentially harmful compounds. One of the most interesting challenges in taste research is to understand how evolution shaped bitter taste receptors to accomplish this task.” [20]

“The universal character of the rejection provoked by intensely bitter molecules demonstrates the close connection between taste and disgust. Toxic compounds, such as strychnine and other alkaloids common among plants, often exhibit a strong bitter taste. In fact, many plants have evolved such compounds to protect themselves from herbivores. All animals generally reject substances with a sour or bitter taste.” [21]

“Human bitter taste receptors of the TAS2R gene family play a crucial role as warning sensors against the ingestion of toxic food compounds.” [52].

“In contrast to the sweet and umami taste, the bitter taste is a signal that prevents animals from ingesting toxic substances.” [53]

“Taste receptors are located on the epithelial surface throughout the alimentary canal to identify nutrients and potential toxins. In the oral cavity, the role of taste is to encourage or discourage ingestion.” [54]

Of course, many bitter substances have a noticeable harmful effect. Reports of plants that are toxic to humans are extensive. However, there is no complete inventory to recognize which contain bitter principles or whether their toxicity is caused by these principles [55]. Most of the poisons used by humans for hunting animals [56], or for intentional poisoning of enemies or suicide, come from bitter plants. Classic examples from history are strychnine (*Strychnos nux vomica*), hemlock (*Conium maculatum*), and aconite (*Aconitum napellus*).

However, many foods, medicines, and additives are bitter or emit bitter tones, which makes their acceptance and consumption difficult. Indeed, the most popular stimulant beverages with caffeine are bitter because of their methylxanthine content. Also, cruciferous vegetables such as broccoli, cauliflower, cabbage, and watercress contain isothiocynates [4]. In many other foods, a certain bitter taste is perceived: olive oil due to compounds derived from oleuropein and ligstroside (from unripe olive fruits); carrots, rich in phenolic acids; most citrus fruits such as oranges, lemons, and grapefruit, as they contain compounds such as naringin and limonin; soy sauce due to the presence of aglycones; grape juices and wines due to the presence of tannins that give them their peculiar flavor; beer, made from the fermentation of malt, where the so-called isohumulones are produced; and most distilled alcoholic beverages, because they contain the compound called 6-n-propylthiouracil (PROP), responsible for their bitterness [57].

On the other hand, many common pharmaceutical compounds have a bitter taste, including antimalarials such as quinine and chloroquine, antibiotics such as erythromycin and floxacins [58], and psychotropic drugs such as haloperidol and procainamide [59]. In fact, most drugs are considered to have a bitter taste [60]. Some of the additives used for food preservation, which are often effective and inexpensive, also produce some unpleasant bitter taste. Generally speaking, chemical compounds that are within the group exhibiting antioxidant potential such as polyphenols, anthocyanins, and carotenoids are responsible for bitter taste [57].

For this reason, one of the purposes of food and pharmaceutical industries is to eliminate or diminish the bitter taste of these beverages, foods, medicines, and additives, either to facilitate their consumption or to avoid the undesirable effects that bitter substances tend to produce. Perhaps the simplest and most traditional formula has been the addition of sugar or sweeteners. In almost all cultures, culinary traditions usually add minimal amounts of seasonings and flavorings to provoke a harmonious balance of flavors, thus achieving, among other intentions, a reduction in the bitter taste. In recent years, the production of attractive foods has been widely explored by nutritional science to avoid unpleasant flavors, including bitterness, astringency, and pungency in various foods [57]. There is also a demand to improve the taste quality of pet and farm animal foods and to develop non-lethal wildlife repellents, e.g., non-toxic chemicals with off-flavors [43]. More recently, with the knowledge of the structure of bitter receptors, the development of bitter blockers is supposed to help mask the bitter taste of drugs [58].

However, although a bitter taste is usually unpleasant and repulsive when it is strong, it is bearable when it is weak [20]. Although bitter foods or beverages are unattractive, in some cases, the bitter taste forms an essential part of the desirable characteristics in that food. Indeed, this is the case with beer, extra virgin olive oil, wines, and some distilled spirits. Another example within beverages is tonic water containing quinine, which is considered a bitter substance tolerable at low concentrations [57]. Therefore, not all bitter compounds are toxic, as they can also be found in high concentrations and with health benefits in foods of vegetal origin [61].

As has been seen, just like with the sweet taste, there is no common chemical structure in the chemicals involved in the perception of bitter taste [21]. Thus, according to Villegas et al., a bitter taste cannot be predicted solely by the structure of the compound molecule. It depends on the size of that molecule, the functional group present, the position of the sugar if present, the decrease in hydrophobicity, and the molecular stereochemistry [57]. A very important characteristic of the bitter sensation is that it can be elicited by very small concentrations of substances, compared to the other tastes [50].

In a way, the discovery of specific receptors for the transduction of bitter taste in recent years has triggered an explosion of research that has not yet come to an end. Despite similar findings for sweet, umami, salty, and sour tastes, the scientific literature is more abundant in publications referring to T2R. Identified in 1996 [21], associated shortly after with T1R receptors, responsible for sweet and umami tastes [44], the genes encoding these receptor proteins were finally discovered in 2000 [45], and by 2006, the genetic map of the T1R and T2R receptors was fully established [47]. Recent genetic studies mention that the ability of humans to detect this taste is determined by a locus on chromosome 5 and possibly chromosome 7 [57]. However, it is known that some bitter compounds can also interact with proton channels [26] and with TRP [43], in particular the TRPm5 channel abundantly present in taste receptor cells [36].

Between 24 and 29 T2R receptors have been identified [62], which raises the question of how the wide range of bitter compounds is that can be detected by such a limited number of sensors [49]. Some suggest that while they all respond to bitter compounds, they cannot necessarily distinguish them from each other, so they would act more like universal bitter sensors [53]. Others, on the contrary, suppose that the basis for discriminating between different bitter compounds lies in the overlap of these receptors on taste papillae, together with their genetic polymorphism, thus allowing humans and animals to detect the enormous range of bitter compounds [23].

However, as with sweet stimuli, because of their large molecular size, bitter substances do not enter the taste cell and bind to the T2Rs [43]. Thus, they activate the G protein gustducin, which is cleaved, and its subunits in turn activate a nearby enzyme that converts precursor molecules inside into what are known as second messengers [24], increasing the intracellular calcium concentration (Ca^2+^), although the mechanism by which this signaling pathway generates the electrical response and the changes that lead to the release of the neurotransmitter that allows signal propagation through the nervous system remains unclear (see Figure 4). Recent studies propose that calcium release stimulates TRPm5 channel receptors to subsequently release ATP, thus suggesting that both T2R and TRP receptors are simultaneously involved in the final transduction of bitter taste [53]. It is unclear why dual signaling should be required for bitter taste and whether it is more than parallel amplification [20].

T2R expression was initially thought to be limited to the oral cavity but has recently been described in a number of other tissues, including the heart, intestine, nasal cavity, and lungs, and other cell types (smooth muscle, endothelial, epithelial, and inflammatory cells) [62]. The collection of gene expression omnibus indicates that T2Rs are expressed in additional human tissues, including the brain, skeletal muscle, endometrium, liver, and omental adipose tissue [59], although its role is still uncertain [61]. It is even remarkable that there are bitter compounds effective via the bloodstream. For example, by injecting a choline solution into the bloodstream of a subject, a bitter taste is suddenly perceived, a sensation which also disappears rapidly [50].

Some suggest that this broad expression of extraoral T2Rs would explain numerous side effects of current drugs, as most of them have a bitter taste [60]. This would imply that because of their bitter character, such drugs produce undesirable effects outside their therapeutic target, so the identification of T2R blockers (antagonists and inverse agonists) would help to eliminate these effects [58]. However, it also means that, in addition to their role in taste transduction, T2Rs in other pathways not only have a deleterious but also a physiological role [25].

The biology of bitter perception is still poorly supported. Neither the sensory receptor cells nor the receptor molecules have been physiologically or molecularly fully defined [45]. Given the amount of work that needs to be undertaken to understand the structure and function of these receptors, the next few years will be an interesting and exciting phase in T2R biology [31].

## 6. The Physiology of Taste and Health

The brain has the capacity to store information associated with what is consumed, which has a direct impact on the attitude and frequency with which certain foods are eaten for the benefit of our health [27]. Applied taste research is increasingly focused on the relationship between diet and health and on understanding the role of the sense of taste in encouraging or discouraging consumption [63].

Sensory information from taste cells is clear to help us detect and respond appropriately to our nutritional needs. The sweet taste of sugars, for example, enhances carbohydrate intake. These taste signals also trigger physiological responses, such as the release of insulin, which facilitates effective utilization of ingested nutrients. Faced with a lack of sodium, animals and humans seek sources of sodium in their intake. According to experimental results, humans and animals with dietary deficiencies tend to ingest foods high in vitamins and minerals [21]. It has also been established that pleasurable foods elevate dopamine levels, which may be crucial for food preferences and food intake control [4].

Current lifestyles are characterized by low physical activity and chronic poor nutrition which predisposes one to metabolic syndrome and its long-term consequences (hypertension, diabetes, and neurodegenerative diseases). The current trend for its treatment involves, among others, the increasing knowledge about the physiology of the sensory systems involved and eating habits. Prenatal experience may have a direct influence on adult food intake choices and could contribute to childhood obesity. Several studies suggest that early exposure to fruits and vegetables or foods high in energy, sugar, or fat have been linked to infants’ preference to consume them. Thus, it may be useful to address early taste perception, how it influences eating behaviors, and to determine its repercussions on lifestyles and human health [4].

In addition, a critical gap in our understanding of taste is how taste mechanisms are related to mood, appetite, obesity, and satiety. The obvious link is that taste guides and largely determines food selection, with salty, sweet, and fatty tastes being the main players. A fascinating link between appetite and mood is that serotonin-enhancing drugs, commonly used to treat mood disorders and depression, have been shown to influence taste thresholds [23].

On the other hand, the available data provide some examples of the role of genetic variation in taste receptors in human nutrition and health. As a result, it may potentially predispose individuals to certain diseases. Thus, some taste receptor alleles may be risk factors for disease. Genotypes of these receptors may be useful as biomarkers to identify predispositions to some diseases and suggest interventions for their prevention [43].

PROP taster status has been found to be associated with a wide range of attributes/phenotypes, including increased perceived orosensory sensations, motion sickness, food preference, fat perception, body mass index, child height, alcohol consumption, and alcoholism [25]. Some studies have shown that supertasters and average tasters of PROP are more discriminating of fat content than nontasters of PROP, which influences their consumption. Indeed, one study showed that nontasters and medium tasters of PROP have a higher body mass index than supertasters in subjects with low dietary constraints. In addition, PROP taster status was found to be associated with stature and choice of eating in children. One study showed that girls with the PROP supertaster had significantly higher weight/height percentiles, whereas boys with the PROP supertaster had significantly lower weight/height percentiles compared with their counterparts. Thus, there is significant evidence of both association and lack of association between PROP taster status and dietary habits, food preferences, and body mass index, leading to the unanswered question of whether genetic variations are a deciding factor in a person’s food habits and preferences [25].

On the other hand, genetic variation in taste receptors may be a biomarker of predisposition to alcoholism. Ethanol has bitter and sweet taste components. Variation in bitter and sweet taste response is associated with ethanol taste perception and beverage alcohol consumption. In mice, allelic variation in the sweet taste receptor gene TAS1R3 is associated with voluntary ethanol consumption. Although hedonic responses to sweet taste are considered one of the biomarkers of predisposition to alcoholism in humans, the genes responsible for this association are still unknown. An increased sensitivity to ethanol bitterness may protect against excessive alcohol consumption. According to this hypothesis, individuals carrying one or two sensitive alleles PAV (dominant haplotype) of the phenylthiocarbamide receptor gene, TAS2R38, had a lower annual consumption of alcoholic beverages than individuals homozygous for the insensitive allele AVI (recessive haplotype). Similarly, there is an association between the risk of alcohol dependence and TAS2R16 polymorphisms: an ancestral allele that is less sensitive to β-glucopyranosides in vitro is associated with an increased risk of alcohol dependence [43]. Other studies have confirmed that PROP taster status is linked to alcohol consumption and alcoholism as a result of the bitter taste it carries [25].

Many cruciferous vegetables are known to contain PROP-like compounds. These compounds are known as glucosinolates and are found in members of the *Brassica*, *Raphanus*, and *Nasturtium* genera, including radishes, cabbages, and watercress. Specific TAS2R38 genotypes are also associated with the perceived bitterness of these glucosinolate-producing plants. Glucosinolates can be toxic to the thyroid gland, so they must be detected before ingestion to prevent thyroid-related diseases. Also, a significant association was found between PROP taster status and vector-induced motion sickness. It was found that PROP nontasters experienced significantly higher degrees of motion sickness and nausea compared to average and super PROP tasters. The authors suggested that susceptibility to motion sickness and nausea has coevolved with nontasters to maintain protection against potential toxin consumption while allowing for the consumption of bitter compounds such as phytonutrients without initial rejection in the oral cavity. This suggests that the nontaster status of PROP is a positive evolutionary selection mechanism [25].

Taste aversion learning, whether natural or experimentally induced, can result from an association of substance and disease, even if there is a gap of several hours between the two. One of the side effects of radiation and chemotherapy treatments in cancer patients is loss of appetite. To a large extent, this is due more to conditioned taste aversions than to the gastrointestinal discomfort produced by these treatments [21].

## 7. Medicinal Bitters

A bitter taste has been shown to promote, at appropriate concentrations, salivation and increased appetite [50]. It has even been known that people who have increased perception to bitter tastes may have a greater overall acuity of the sense of taste and increased sensitivity to olfactory signals. In addition, it has been noted that many of the chemicals that impart a certain bitter taste to foods may be beneficial to health. Polyphenols, methylxanthines, isoflavones, and sulfonamides, present in cruciferous vegetables and carrots, have antioxidant effects and protect us from certain diseases [4]. Tannins present in grape wine may have a cardiovascular protective effect [64]. Xanthines contained in stimulant beverages, in addition to their stimulant effect, offer benefits at the cerebrovascular level and even serve to control migraines [65]. Therefore, despite not being entirely pleasant, their consumption is compensated by the health benefits of these unpleasant-tasting compounds [57].

On the other hand, an interesting chapter opens with the understanding of the role of T2Rs responsible for the detection of bitter taste, playing an important role in the maintenance of homeostasis and in the response to certain diseases [59]. Accumulating evidence suggests that T2R-mediated signaling contributes greatly to innate immunity in organ epithelia that are connected to the external environment [60]. A recent study confirmed that T2Rs in human airways detect secreted bacterial products, stimulating a secretion of antimicrobial peptides into the mucosa capable of inhibiting a variety of respiratory pathogens [66]. Another study showed that at the pulmonary level, T2R agonists increase the frequency of ciliary movements in epithelial cells, induce bronchial smooth muscle cells to relax, and modulate the production of pro-inflammatory mediators [62].

There is also increasing evidence of the effect of T2R activation at the gut level on epithelial cells and endocrine cells, promoting the secretion of hormones such as incretin (glucagon-like peptide-1), ghrelin, and other substances such as cholecystokinin and glucose-dependent insulinotropic polypeptide (GIP) [67], with obvious physiological and possible therapeutic effects. The existence of these “taste” mechanisms in the gut is perhaps not surprising, given the importance of sensing the chemical nature of light contents at all points along the gastrointestinal tract. However, the findings have generated new excitement in understanding how the gut participates in the sensing and control of appetite in general and digestive processes in particular [23].

Many foods and medicinal plants contain polyphenols, which are bitter-tasting substances which have shown therapeutic benefits. Recent studies confirm that polyphenols are ligands for extraoral T2Rs and this interaction in T2Rs at the brain level helps regulate appetite/satiety control, while T2Rs at the gut level contribute to the metabolism of adipose tissue and promote the secretion of gastrointestinal hormones such as ghrelin, which helps control blood glucose. “Together, these factors can contribute to understand the beneficial effects of polyphenol-rich diets but also might aid in identifying new pharmacological pathway targets for the treatment of diabetes and obesity.” [9].

Deshmukh et al, summarized this as follows: “The plants containing bitters (bitter substances) are exclusively used due to their intensively bitter tastes, they increase the appetite and stimulates digestion by acting on the mucous membranes of the mouth. Also increases the flow bile, stimulate repair of gut wall lining and regulate the secretion of insulin and glucagon. They reflectively induce stimulation on the salivary gland and the secretion of gastric juice. The bitter taste recognizes at the very moment of excitation. The taste buds are situated at the root of the tongue and keep the bitter taste some time afterwards (‘after-taste’). These are then sending the signal to the brain, which by reflexes than affect the swelling of the secreting mucous membranes. The bitters have therefore positive influence on the digestive processes, then it led to an improvement on appetite as well as an amelioration of the food intake. The bitter plants are generally showing antidiabetic and tonic action.” [68].

It has even been claimed that any T2R could mediate unanticipated physiological changes in any T2R-expressing tissue exposed to significant concentrations of a bitter-tasting drug. T2Rs in respiratory and gastrointestinal tissues would be particularly available for ingestion or inhalation, but drugs absorbed into the bloodstream could reach T2Rs in almost any tissue in the body. This hypothesis, if confirmed, would have important implications for drug development and medical practice. Bitter-tasting phytochemicals, some of which are suspected to have biological activity, could similarly exert their therapeutic actions through extraoral T2Rs if target concentrations were sufficiently high [59].

As mentioned above, bitter-tasting molecular structures are not exclusive to a single type of secondary metabolite. Indeed, bitter principles are found in alkaloids, heterosides, essential oils, terpenes, isoprenoids, tannins, mucilages, alcohols, acids, and esters. For this reason, we must consider that more than the molecular structure, their bitter character activates their physiological and perhaps therapeutic function at oral and extraoral T2Rs. From a chemistry point of view, we still need a scientific explanation to determine exactly what happens at the molecular level to make some of these secondary metabolites bitter and others not. Table 2 shows some bitter compounds from food or medicinal plants with therapeutic effects.

Despite this novel evidence, phytochemical and pharmacological research focused on exploring the therapeutic role of medicinal plants is still in its incipient stages, not because of their chemical compounds per se but because these compounds have a bitter taste. Using a statistical inquiry, Grădinaru et al. offer a pioneering study on the relevance of phytochemical taste for anti-inflammatory and anti-cancer activity. Out of 624 phytochemicals, 461 were found to possess anti-cancer activity, and they found a robust statistical association between the bitter taste of these phytochemicals and their simultaneous anti-cancer and anti-inflammatory effects [69].

Finally, Dragoș et al. affirm: “The anti-inflammatory activity of plant-derived compounds is more strongly associated with their taste than with their chemical class. Bitter phytocompounds have a higher probability of playing an inflammation related role, exerting anti-inflammatory activity, by contrast to sour and sweet phytochemicals, which have a higher probability of being devoid thereof.” [70].

## 8. Conclusions

For many years, the theory of taste physiology considered that the function of identifying bitter tastes was to detect and prevent the consumption of toxins from plants, decaying foods, and drugs. In my experience with indigenous peoples, I was surprised to learn that for them bitter medicinal plants have important medicinal, preventive, and therapeutic value. In reviewing recent scientific advances, we found that the T2R membrane receptors for bitter taste, G protein-associated or metabotropic receptors, are not only located in the oral mucosa but are also found in other organs and tissues of the human being, also fulfilling an important physiological and therapeutic role.

Indeed, several studies have confirmed their antioxidant effects, anti-inflammatory activity, appetite/satiety control, cardiovascular protection, increased frequency of ciliary movements in epithelial cells, maintenance of homeostasis, production of pro-inflammatory mediators and stimulation of antimicrobial peptide secretion, among others, with potential benefits for the prevention and treatment of many diseases.

Thus, an important chapter of research on bitter T2R receptors and new epistemological and research routes for medicinal plants has been opened. Perhaps the indigenous people are right. With this new knowledge, we will be able to understand the true biological basis through which bitter plants may act as “good medicine” [60].

## Figures and Tables

**Figure 1 pharmaceuticals-17-00722-f001:**
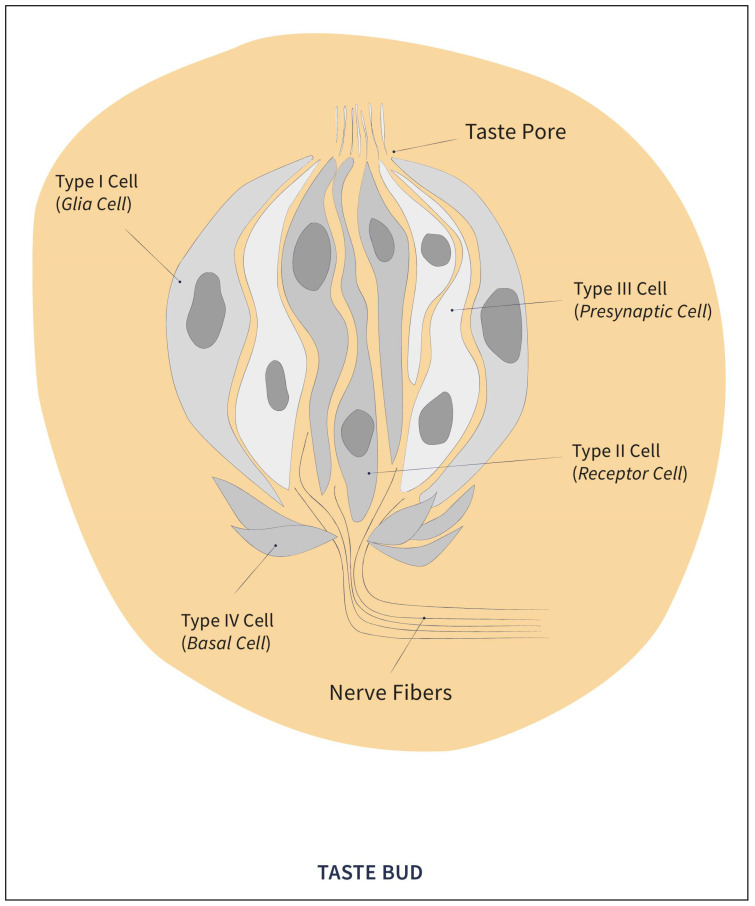
Structure of the taste button.

**Figure 2 pharmaceuticals-17-00722-f002:**
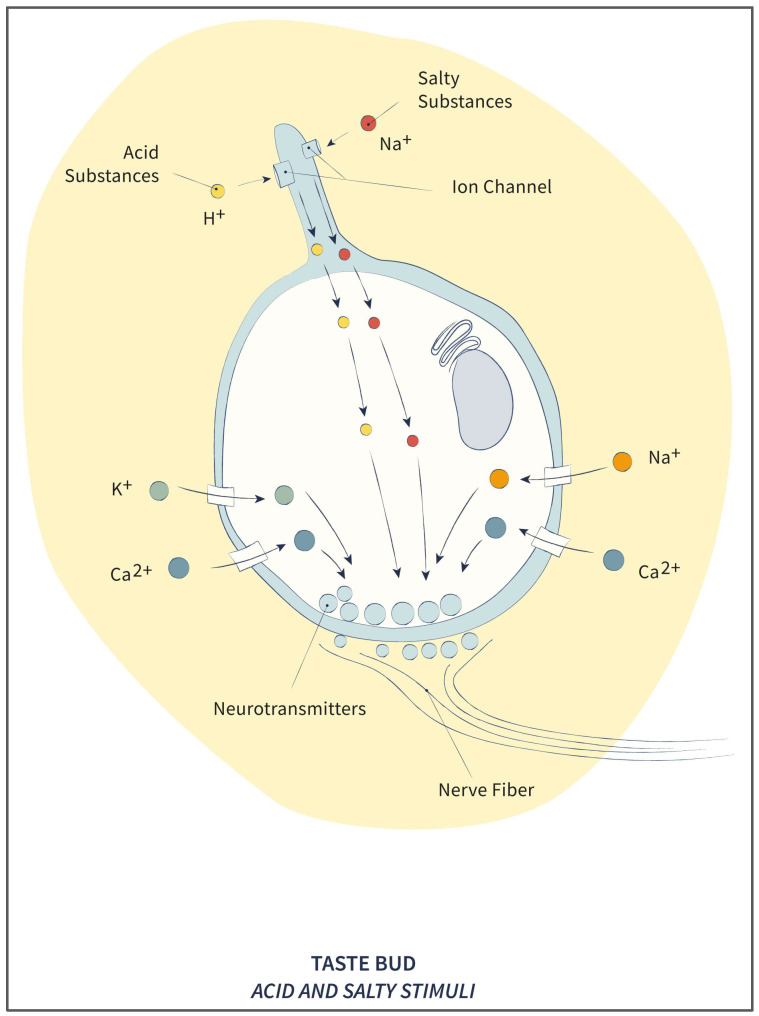
Scheme of the cellular mechanism for the transduction of sour and salty tastes.

**Figure 3 pharmaceuticals-17-00722-f003:**
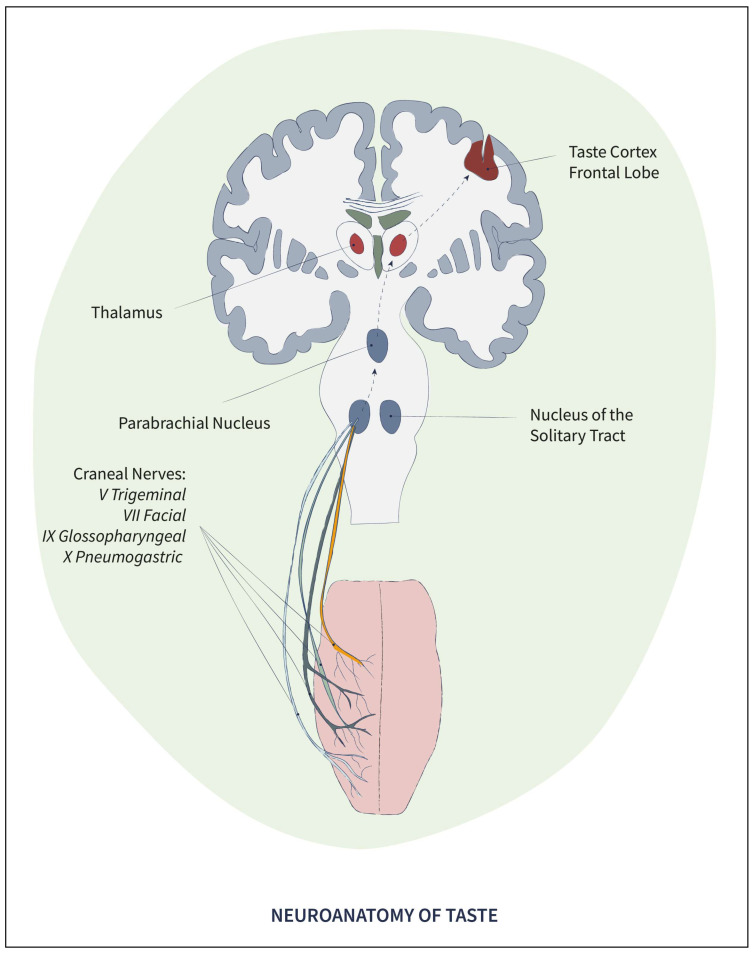
Scheme of the physiological pathway of flavors.

**Figure 4 pharmaceuticals-17-00722-f004:**
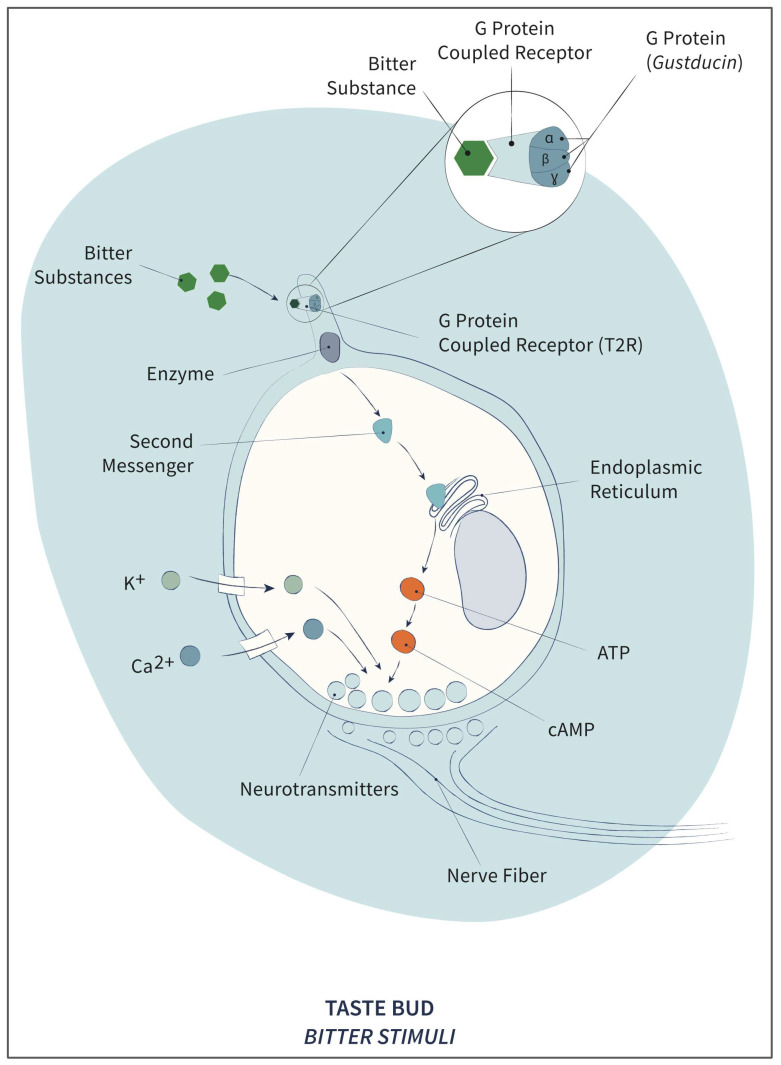
Scheme of the cellular mechanism for the transduction of bitter taste.

**Table 1 pharmaceuticals-17-00722-t001:** Bitter medicinal plants reported from cosmopolitan or local communities.

Scientific Name	Common Name	Bitter Compounds	Distribution	Ref.
*Artemisia absinthium*	Ajenjo (span.), Wormwood (eng.)	Thujone, sesquiterpene lactones: absinthin, anabsinthin, anabsin, and artabsin	Cosmopolitan	[10]
*Artemisia vulgaris*	Altamisa (span.), Mugwort (eng.)	Terpenoids and sesquiterpene lactones	Cosmopolitan	[11]
*Azadirachta indica*	Nim (span.), Neem (eng.)	Azadirachtin, meliacin, gedunin, nimbidin, nimbolides, salanin, nimbin, valassin, meliacin, limonoid	Indian	[12]
*Banisteriopsis caapi*	Ayahuasca, yage, capi (span. and indigenous languages)	β-carboline alkaloids	Amazon region	[7]
*Coptidis rhizoma*	Huang Lian (Chinese)	Alkaloids, lignans, phenylpropanoids, flavonoids, phenolic compounds, and steroids	Traditional Chinese medicine, Japanese traditional herbal medicine (Kampo)	[13]
*Garcinia kola*	Cola amarga (span.), Orogbo (Yoriba)	Biflavonoids, benzophenones, benzofurans, benzopyran, xanthones, and phytosterols	Traditional African medicine	[14]
*Gentiana purpurea*	Genciana (span.), Purple Gentian (eng.)	Secoiridoids, amarogentin	Ancient European tradition	[15]
*Momordica charantia*	Melón amargo (span.), Bitter melon (eng.)	Phenolic components, Kuguaglycoside G, Momordicine	Traditional Chinese medicine	[16]
*Phellodendron amurense bark*	Amur cork tree (eng.), Huáng bò (Chinese)	Phellodendrine, Berberine	Traditional Chinese medicine	[17]
*Verbena officinalis*	Verbena (span.), Vervain (eng.)	Iridoids, flavonoids and phenolic acid derivatives	Cosmopolitan	[18]
*Vernonia amygdalina*	Hoja amarga (span.), Ewuro (Yoruba)	Sesquiterpene lactones: vernodalin, vernolide, hydroxyvernolide	Tongwe (Tanzania)	[19]

**Table 2 pharmaceuticals-17-00722-t002:** Bitter compounds of food or medicinal plants con therapeutic effect.

Bitter Compounds	Food or Medicinal Plants	Therapeutic Effect	Ref.
Polyphenols, methylxanthines, isoflavones, and sulfonamides	Cruciferous vegetables and carrots	Antioxidant effects and protect us from certain diseases	[4]
Tannins	Grape wine	Cardiovascular protective effect	[64]
Xanthines	Stimulant beverages (coffee, tea, chocolate, kola)	Benefits at the cerebrovascular level and control migraine	[65]
Polyphenols	Exist widely in plant and plant-based foods	Metabolism of adipose tissue and promotes the secretion of gastrointestinal hormones such as ghrelin, which helps control blood glucose	[9]
Thujone, sesquiterpene lactones: absinthin, anabsinthin, anabsin, and artabsin	Wormwood (*Artemisia absinthium*)	Anti-inflammatory, antioxidant, and antimicrobial effects, and it has been shown to influence GI tract and urinary system	[10]
Azadirachtin, meliacin, gedunin, nimbidin, nimbolides, salanin, nimbin, valassin, meliacin, limonoid	Neem (*Azadirachta indica*)	Antiplasmodial, antitrypanosomal, antioxidant, anti-cancer, antibacterial, antiviral, larvicidal, and fungicidal activities.	[12]
Phenolic components, Kuguaglycoside G, Momordicine	Bitter melon (*Momordica charantia*)	Antidiabetic, anti-obesity, anti-inflammatory, anti-cancer effects among others	[16]
Phellodendrine, Berberine	Amur cork tree (*Phellodendron amurense*)	Protect against airway inflammation, can reduce blood uric acid levels, reduce blood glucose levels, and slow the development of diabetic nephropathy	[17]
Sesquiterpene lactones: vernodalin, vernolide, hydroxyvernolide	Bitter leaf (*Vernonia amygdalina*)	Alleviate inflammation, pyrexia and nociception	[19]

## Data Availability

Data sharing is not applicable.

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
