# Peer review of "Potential of Bitter Medicinal Plants: A Review of Flavor Physiology"

_pharmaceuticals, 2024, doi:10.3390/ph17060722_

Round 1

Reviewer 1 Report

Comments and Suggestions for Authors

The manuscript topics "better test in medicinal plants" is very interesting. However, there are some issues  before recommending the manuscript for publication.

1. The paper suffers from some inadequate in presentation. No figure or table were provided in the manuscript to compare better test in medicinal plants.

2. The introduction is not sufficient. The author is encourage to embed more contents in introduction regarding to the study significance, objectives and problem stetmente.

Reviewer 2 Report

Comments and Suggestions for Authors

Dear Authors,

The Manuscript ID pharmaceuticals-3020313, Titled “POTENTIAL OF BITTER MEDICINAL PLANTS: A REVIEW OF FLAVOR PHYSIOLOGY” should be progressed for better scientific worth. The novelty of this review is low and the manuscript has some problems of data presented.

There are numbers of shortcomings.

The Abstract summarizes the main findings but it is not well-structured. Addionally, the purpose of the study is not clearly highlighted; the methods are not described, nor are the databases searched through. The section is written in narrative style which should be avoided; scientific language should be followed. This section should not exceed 200 words. Generally, the abstract should be rewritten and optimized according to the requirements.

The Introduction should define the object of the work and its worth, including specific hypotheses being tested. Herein, the presented state of the research is not reviewed carefully and key publications were not cited. There are personal statements not supported by proper literature and more appropriate for the Discussion section (Lines 64-102). Finally, the citations are not done according to the requirements.

Materials and Methods: databases which were searched to identify research papers should be written and appropriately cited.

Results does not provide a concise and precise description of the data, their interpretation as well as the experimental conclusions that can be drawn. Also, some of the data from the literature survey should be presented in tables, e.g., an overview of the main classes of secondary metabolites with bitter taste and etc. Thus, the review would be completed.

Finally, the study should be reorganized and rewritten for better scientific soundness and interest to reader.

In conclusion, based on the evaluation of its originality, significance of content, quality of the presentation, scientific soundness, and interest to readers, after considering the above mentioned comments and suggestions, I recommend Manuscript ID: pharmaceuticals-3020313  for “Reject”  in Pharmaceuticals.

Comments on the Quality of English Language

Minor editing of English language required.

Reviewer 3 Report

Comments and Suggestions for Authors

Based on the title and abstract of the manuscript I expected a review report on medicinal plants with bitter taste, providing information on active compounds and physiological effects, pharmacology and therapeutic potential of their bitter-tasting compounds. Unfortunately, these aspects are largely missing from the manuscript.

The topic suggested by the second part of the title "flavor physiology" is covered by the manuscript, however, even this review is not based on the most recent sources available. Most cited references are 20 years old or even earlier, and even when the author claims "the most recent and updated publications" (Line 451 and below) the most recent source is from 2001! Thus the text in many places sounds rather like excerpts from a textbook, not the synthesis of the most up-to-date research outcomes, as one would expect from a review paper.

In addition, there is lot of redundancy in the text, i.e. the same information is repeated in multiple places, which is unnecessary, e.g. stimulant beverages like coffee and tea are mentioned several times at different places, similarly to polyphenols, citrus fruits, etc.

Since the topic is interesting, I would recommend the author thoroughly revises the manuscript, which should:

- summarize the main points of related literature sources from the last couple of years (from 2020 onwards)

- present several examples of bitter tasting medicinal plants (focusing or at least including the traditional healing practice of the Inga Indians) - together with physiologically active compounds and their health benefits.

Just a few recommendations on 2024 literature sources related to this topic:

Marta Trius-Soler, Juan José Moreno (2024) Bitter taste receptors: Key target to understand the effects of polyphenols on glucose and body weight homeostasis. Pathophysiological and pharmacological implications, Biochemical Pharmacology, 2024, 116192, https://doi.org/10.1016/j.bcp.2024.116192.

Zeyu Zhao, Fang Song, Shunsuke Kimura, Takeshi Onodera, Takahiro Uchida, Kiyoshi Toko (2024) Taste sensor for detecting non-charged bitter substances: Xanthine derivatives of pharmaceutical applications, Microchemical Journal, Volume 200, 2024, 110248, https://doi.org/10.1016/j.microc.2024.110248.

Håvard Hoel, Hugo J. de Boer, Anneleen Kool, Helle Wangensteen (2024) Analysis of bitter compounds in traditional preparations of Gentiana purpurea L, Fitoterapia, Volume 175, 2024, 105932, https://doi.org/10.1016/j.fitote.2024.105932.

Reviewer 4 Report

Comments and Suggestions for Authors

I reviewed the article "POTENTIAL OF BITTER MEDICINAL PLANTS: A REVIEW OF FLAVOR PHYSIOLOGY" by Zuluaga Germán.

This review aims to understand the relationship between the bitter taste of certain foods and their possible therapeutic or, conversely, toxic effect.

The collection is fascinating, well-written, and well-proposed. For this reason, I congratulate the author. I have only a few suggestions to improve the style and resonance of the manuscript.

·       I would suggest that the author add figures to guide the reader to an immediate understanding of the text. For example, regarding the signal transduction mechanism of bitter taste, highlighting the receptors involved.

·       Could the author add a small paragraph about intraspecific differences in bitter taste perception?

·       Lines 495-497: In this regard, I suggest the author add a table containing some examples of bitter-tasting foods and their therapeutic effects. For example, bergamot (Citrus bergamia) has a very strong bitter taste due to the presence of naringin, but it has extraordinary antiaging, antioxidant, and antidiabetic properties (doi: 10.3389/fphys.2023.1225552). Similarly, bitter melon (Momordica charantia) has a bitter taste due to the presence of several triterpenoids, but it has antidiabetic, anti-obesity, anti-inflammatory, and anticancer properties (10.1016/j.ijbiomac.2023.123173). I suggest including these examples as well.

·       I suggest the author also provide some examples of toxic food with a bitter taste.

Round 2

Reviewer 2 Report

Comments and Suggestions for Authors

Dear Author,

An enhancement of the manuscript is completed but a minor revision should be done on the revised Manuscript ID:  pharmaceuticals-3020313.

The remarks and proposals previously given were considered. An improvement of the manuscript is done. Nevertheless, concerning the Table 1, the chemical compounds referred to the bitter taste should be included. Also, column “Traditional village” is better to be changed to “Distribution”.

I should recommend a Graphical abstract to be presented.

Additionally, The Conclusion section should be summarized and gives the highlight of the review.

Comments on the Quality of English Language

 Minor editing of English language required.

Reviewer 3 Report

Comments and Suggestions for Authors

I appreciate the author's effort to improve the manuscript along the lines suggested by reviewers. I accept the answers provided for my critical notes. Inserting a table with examples of bitter medicinal plants was a good solution. Similarly, the figures that now are part of the manuscript contribute to better understanding of taste perception. However, it is not clear for me if these are original figures created by the author or were taken from a secondary source (if the latter is the case, please indicate the source of each figure). Also, the labels in the figures would be easier to read with larger font size.
On the whole, the manuscript has been substantially improved, and can be accepted for publication in my opinion.
